# Comparative Characteristics and Zoonotic Potential of Avian Pathogenic *Escherichia coli* (APEC) Isolates from Chicken and Duck in South Korea

**DOI:** 10.3390/microorganisms9050946

**Published:** 2021-04-27

**Authors:** Jiyeon Jeong, Ji-Youn Lee, Min-Su Kang, Hye-Jin Lee, Seong-Il Kang, O-Mi Lee, Yong-Kuk Kwon, Jin-Hyun Kim

**Affiliations:** Avian Disease Research Division, Animal and Plant Quarantine Agency, 177, Hyeoksin 8-ro, Gimcheon-si 39660, Gyeongsangbuk-do, Korea; jjy11214@snu.ac.kr (J.J.); enteric@korea.kr (J.-Y.L.); kangmskr@korea.kr (M.-S.K.); soulmate368@naver.com (H.-J.L.); ksilion@korea.kr (S.-I.K.); omilee@naver.com (O.-M.L.); kwonyk66@korea.kr (Y.-K.K.)

**Keywords:** avian pathogenic *E. coli*, serogroups, phylogenetic groups, virulence-associated genes, antimicrobial resistance, *mcr-1*, ST95, zoonotic potential, public health, poultry

## Abstract

Avian pathogenic *Escherichia coli* (APEC) causes colibacillosis, which is an economically important disease in the poultry industry worldwide. The present study investigated O-serogroups, phylogenetic groups, antimicrobial resistance, and the existence of virulence-associated genes (VAGs) and antimicrobial resistance genes in 125 APEC isolates between 2018 and 2019 in Korea. The phylogenetic group B2 isolates were confirmed for human-related sequence types (STs) through multi-locus sequence typing (MLST). O-serogroups O2 (12.5%) and O78 (10.3%) and phylogenetic group B1 (36.5%) and A (34.5%) were predominant in chicken and duck isolates, respectively. Out of 14 VAGs, *iucD*, *iroN*, *hlyF*, and *iss* were found significantly more in chicken isolates than duck isolates (*p* < 0.05). The resistance to ampicillin, ceftiofur, ceftriaxone, and gentamicin was higher in chicken isolates than duck isolates (*p* < 0.05). The multidrug resistance (MDR) rates of chicken and duck isolates were 77.1% and 65.5%, respectively. One isolate resistant to colistin (MIC 16 μg/mL) carried *mcr-1*. The B2-ST95 APEC isolates possessed more than 9 VAGs, and most of them were MDR (82.4%). This report is the first to compare the characteristics of APEC isolates from chickens and ducks in Korea and to demonstrate that B2-ST95 isolates circulating in Korea have zoonotic potential and pose a public health risk.

## 1. Introduction

*Escherichia coli*, a normal inhabitant in the intestine, can be pathogenic and cause a spectrum of diseases in both humans and animals [1]. Pathogenic *E. coli* is classified into two main categories on the basis of the host site of colonization and potential progression to infection: diarrheagenic *E. coli* (DEC), which cause enteric infections, and extraintestinal pathogenic *E. coli* (ExPEC), which cause infections outside of the gastrointestinal tract [2,3,4]. In particular, avian pathogenic *E. coli* (APEC), a subtype of ExPEC, is the main causal agent of colibacillosis in poultry farms, including respiratory tract infection, septicemia, cellulitis, and polyserositis [5]. Thus, APEC is a major cause of economic losses in the poultry industry worldwide due to high morbidity and mortality in poultry flocks [6].

Therefore, for effective surveillance and proper prevention of colibacillosis, it is important to understand the phylogeny, lineage, and virulence of APEC strains that are prevalent in poultry farms. APEC typing techniques include serotyping, plasmid profiling, phylogenetic typing, pulsed-field gel electrophoresis (PFGE), multi-locus sequence typing (MLST), whole-genome sequencing (WGS), etc. [1]. Of these, O-serogrouping relies on the detection of the lipopolysaccharide (somatic of O) antigen. Among more than 180 O-serogroups, *E. coli* belonging to serogroups O1, O2, O8, O35, and O78 are most often implicated in colibacillosis [7]. *E. coli* can also be divided into eight phylogenetic groups (*E. coli sensus stricto* A, B1, B2, C, D, E, F, and *E. coli* cryptic clade I) [8]. Logue et al. [9] reported that most human ExPEC were classified into groups B2 and D, whereas a majority of APEC belonged to group C, followed by F, B1, and B2. Other studies have identified various groups, including A, B1, and B2 in APEC isolates [10,11]. MLST is a standard molecular subtyping technique for determining that is used to determine the genetic relatedness among strains and identify the strains with high discriminatory power [12]. In addition, MLST is widely used in human ExPEC and APEC genotyping because their sequence types (STs) can be compared globally based on country of isolation and specific source with the aid of a web-based database (http://enterobase.warwick.ac.uk/species/index/ecoli, accessed on 14 February 2021).

In addition to APEC, uropathogenic *E. coli* (UPEC), neonatal meningitis-causing *E. coli* (NMEC), and septicemia-causing *E. coli* (SEPEC) are types of ExPEC. These pathotypes may cause urinary tract infections (UTIs), neonatal meningitis, and septicemia, respectively, in humans [2,3,4]. A large number of comparative studies between APEC and human ExPEC strains have shown that a notable degree of overlap in serogroups, STs, and virulence-associated genes (VAGs); these results support that APEC is zoonotic in nature [13,14,15].

The pathogenic ability of ExPEC, including APEC, is mediated by various virulence factors, of which VAGs encode a broad range of products such as adhesins, iron-acquisition systems, toxins, and protectins [1]. A genetic analysis study revealed that APEC strains could be a reservoir of VAGs of ExPEC in humans [16]. The *fimC*, *iss*, *tsh*, *hlyF*, and *iroN* genes are predominantly detected in APEC [1,17]; of these, the *iss* gene, which encodes a protein linked to increased serum survival in human *E. coli* isolates, and the *hlyF* gene, encoding hemolysin, are also observed in human ExPEC [18]. Exchange of these VAGs between human ExPEC and APEC strains is possible [18]. Consequently, these APEC strains pose a public health threat through the food supply.

Antimicrobial resistance is also a significant public health risk and makes it difficult to treat bacterial diseases in poultry. However, treatment for colibacillosis relies on the use of antibiotics worldwide [19]. So far, many APEC isolates remain sensitive to several antimicrobials, but a growing number of strains are resistant to antibiotics, and the emergence of multidrug resistance (MDR) strains is accelerating due to excessive antibiotic use [19,20]. In addition, many studies have found that APEC isolates act as a reservoir of antimicrobial resistance genes that could be transmitted to other bacteria by horizontal gene transfer [21,22,23]. Drug-resistant APEC strains and APEC strains harboring antimicrobial resistance genes in poultry may contaminate the food supply from farm to fork through poultry meat and poultry products.

Numerous recent studies on serogroups, phylogenetic groups, and antimicrobial resistance in APEC isolates from chickens have been conducted in many countries [24,25,26,27], but such surveys are rarely performed on APEC isolates from ducks [28,29]. In Korea, the characteristics of APEC isolated from ducks have rarely been investigated as well [30], although consumption of duck meat has recently increased. APEC isolates have zoonotic potential and pose public health risks because they share many important characteristics, including serogroups, STs, and virulence factors, with human ExPEC. In Korea, most prior studies have focused only on the characterization of APEC isolates [17,30,31,32]. Therefore, the purpose of this study was to investigate and compare the serogroups, phylogenetic groups, and genetic characteristics of recent APEC isolates from ducks and chickens in Korea. In addition, the zoonotic potential and public health risk of the APEC isolates corresponding to group B2, which is known as a major group of human ExPEC strains, was assessed.

## 2. Materials and Methods

### 2.1. Bacterial Isolates

We used 125 APEC isolates that are archived by the Avian Disease Research Division at the Animal and Plant Quarantine Agency (APQA) in Korea. The isolates were collected from chickens (96 isolates) and ducks (29 isolates) diagnosed with colibacillosis from 60 chicken farms and 23 duck farms in Korea between 2018 and 2019. Their sources were the liver, infraorbital sinus, articular cavity, air-sac, heart, yolk, inner ear, femoral head, and wattle. The number of isolates according to sample source in chickens and ducks is shown in Appendix A. Presumptive *E. coli* isolates were identified using VITEK 2 Gram-Negative Identification (GNI) cards (bioMérieux, Durham, NC, USA) following the manufacturer’s instructions. If the same antimicrobial susceptibility patterns were shown in several isolates from the same farm, one of these isolates was randomly selected for this study. These isolates were stored at −80 °C in 30% (*v*/*v*) glycerol in Tryptic soy broth (Difco Laboratories, Detroit, MI, USA) until further use.

### 2.2. O-Serogrouping

To identify the serogroup of the APEC isolates, O-serogroup typing was performed by slide agglutination using polyvalent and monovalent antisera for 182 somatic O-serogroups (Joongkyeom, Goyang, Korea). Any isolate that showed a positive reaction with polyvalent antisera was retested with monovalent antisera. *E. coli* isolates were cultured on Tryptic soy agar (Difco Laboratories, Detroit, MI, USA) at 37 °C overnight. Each agar plate suspension, at a concentration of 1.8 × 10^9^ colony forming units (CFU)/mL, was autoclaved at 121 °C for 15 min. After discarding the supernatant, 500 μL of sterilized saline was added to the pellet and mixed for use as the antigen. A total of 20 μL of the antigens were mixed with 20 μL of the antisera on a glass slide. After the slide was hand-tilted for 2 min, the presence and absence of agglutination were read as positive and negative, respectively.

### 2.3. DNA Extraction and Phylogenetic Group Determination

Genomic DNA used for polymerase chain reaction (PCR) analysis was extracted from colonies grown on Tryptic soy agar (Difco Laboratories, Detroit, MI, USA) using the commercial kit DNeasy Blood and Tissue (Qiagen, Hilden, Germany). The phylogenetic groups (A, B1, B2, C, D, E, F, and clade I) of the 125 APEC isolates were determined using multiplex PCRs of the genes *arpA*, *chuA*, *yjaA*, *TspE4.C2*, and *trpA*, as previously described [8]. Representative positive PCR products of each gene were sent for sequencing of both strands (Macrogen, Seoul, South Korea). The sequences were aligned using CLC Main Workbench software (CLC Bio, Aurhus, Denmark), and the aligned sequence was compared with sequences in GenBank^®^ by the BLST program (National Center for Biotechnology Information, Bethesda, MD, USA) to determine the sequence identity.

### 2.4. Detection of Virulence-Associated Genes (VAGs)

The presence of 14 virulence genes in the 125 APEC isolates was determined by PCR, as previously described [17]. The virulence factors that were tested included adhesins (*fimC*, *tsh*), iron acquisition (*iroN*, *irp2*, *iucD*, and *fyuA*), toxins (*lt*, *st*, *stx1*, *stx2*, *vat*, and *hlyF*), and resistance to bactericidal factors (*ompT* and *iss*). The details and size of the amplified products for all primers are listed in Appendix A. Representative positive PCR products of each gene were sent for sequencing of both strands (Macrogen, Seoul, South Korea) and analyzed as described above. As detailed in El-Shaer et al. [33], a mean virulence score (MVS) was calculated as the sum of all VAGs detected in the isolates in a given phylogenetic group divided by the number of isolates in that group. The MVS was used to quantify and compare the retention of VAGs among the isolates belonging to each phylogenetic group.

### 2.5. Antibiotic Susceptibility Test

The minimum inhibitory concentrations (MICs) of the 125 APEC isolates were determined by the broth microdilution method using a Sensititre automated antimicrobial susceptibility system (Trek Diagnostic Systems, East Grinstead, UK) and agar dilution method, according to the manufacturer’s instructions and the protocols of the Clinical and Laboratory Standards Institute, respectively [34]. The following 17 antimicrobials were tested: amoxicillin/clavulanic acid, ampicillin, azithromycin, cefoxitin, ceftiofur, ceftriaxone, chloramphenicol, ciprofloxacin, colistin, doxycycline, enrofloxacin, gentamicin, nalidixic acid, streptomycin, sulfisoxazole, tetracycline, and trimethoprim/sulfamethoxazole. *Escherichia coli* ATCC 25922 and *Staphylococcus aureus* ATCC 29213 were used as quality control strains in MIC determinations. The interpretation criteria of MIC are available in the CLSI, CDC, and EUCAST [34,35,36,37]. Each isolate was categorized as no drug (NDR), non-multidrug (SDR), and multidrug (MDR) resistant, as previously described [38]. SDR was defined as resistance to 1 or 2 of the 10 antimicrobial classes tested in this study. An isolate was considered multidrug resistant (MDR) if it was resistant to at least three different classes of antimicrobials [39].

### 2.6. Detection of Antimicrobial Resistance Genes

Detection of various antimicrobial resistance genes was carried out with a PCR assay. Genes encoding macrolide resistance (*mphA*, *mphB*, *ermA*, *ermB*, *ermC*, *ereA*, *ereB*, *mefA*, and *msrA*) [40], plasmid-mediated quinolone resistance (PMQR) genes (*qnrA*, *qnrB*, *qnrC*, *qnrD*, *qnrS*, *aac6′-1b-cr*, and *qepA*) [41,42,43,44,45], and genes conferring resistant to polymyxin (*mcr1*, *mcr2*, *mcr3*, *mcr4*, and *mcr5*) [46,47], β-lactam penicillin (*bla*_SHV_, *bla*_TEM_, *bla*_CTX-M_ group Ⅰ, *bla*_CTX-M_ group Ⅱ, *bla*_CTX-M_ group Ⅲ, and *bla*_CTX-M_ group Ⅳ) [48,49], phenicol (*cmlA*, *cat*, and *floR*) [50,51], tetracycline (*tetA*, *tetB*, *tetC*, *tetD*, *tetE*, and *tetG*) [52,53,54], aminoglycoside (*strA-B* and *aadA*) [55], and sulfonamide (*sul1* and *sul2*) [56] were detected using previously described primers and protocols. Isolates resistant to each antibiotic were screened for antimicrobial resistance genes associated with each antibiotic. The PCR products of the representative samples were sequenced (Macrogen, Seoul, South Korea) and analyzed as described above.

### 2.7. Multi-Locus Sequence Typing (MLST) Analysis

A total of 22 isolates belonging to phylogenetic group B2 were subjected to MLST analysis to delineate their clonal relationships. MLST was performed based on seven housekeeping genes (*adk*, *fumC*, *gyrB*, *icd*, *mdh*, *purA*, and *recA*), as described previously [57]. The allelic number and corresponding sequencing type (ST) number were designated according to the scheme on the *E. coli* MLST website (http://enterobase.warwick.ac.uk/species/index/ecoli, accessed on 14 February 2021). A dendrogram for visualizing genomic relations among 22 isolates was generated using BioNumerics v. 6.0 (Applied Maths, Sint-Martens-Latem, Belgium) based on joint analysis of the seven housekeeping genes.

### 2.8. Statistical Analysis

The frequencies of antimicrobial resistance, virulence genes, and phylogenetic groups between APEC isolates from chickens and ducks were statistically compared by the Chi-squared test and Fisher’s exact test using SigmaPlot 14.0 (Systat Software Inc., San Jose, CA, USA). The *p*-value was calculated and considered significant below α = 0.05.

## 3. Results

### 3.1. O-Serogroup Distribution of APEC Isolates

The O-serogroup distribution of the APEC isolates is presented in Table 1. Among all 125 APEC isolates, 86 (68.8%) were classified into 39 O-serogroups. However, 39 isolates (31.2%) were non-typable. The most prevalent serogroup was O2 (10.4%), followed by O78 (5.6%), O1 (5.6%), O45 (4.8%), and O88 (4.0%). The most frequent serogroups of the 96 chicken isolates were O2 (12.5%), O1 (6.3%), O45 (6.3%), and O88 (5.2%), whereas those of the 29 duck isolates were O78 (10.3%), followed by O15, O25, and O150 (6.9% each). In particular, serogroup O2, which was most commonly found in chicken isolates, was detected in all breeds of chickens (layer 4/22, broiler 4/41, broiler breeder 1/17, Korean native chicken 3/16) (Appendix A).

### 3.2. Phylogenetic Groups and VAGs

The distribution of phylogenetic groups among the 125 APEC isolates is presented in Figure 1. Among all 125 APEC isolates, phylogenetic group B1 (35.2%) was the most common, followed by group B2 (17.6%), group F (16.8%), group A (12.8%), group E (10.4%), and group C (7.2%). For the 96 chicken isolates, group B1 (36.5%) and group B2 (20.8%) were predominant, whereas, in the 29 duck isolates, group A (34.5%) and group B1 (31.0%) were mainly prevalent. Group A was significantly more frequent in the isolates from ducks compared to chickens (*p* < 0.05). The prevalence of VAGs in the 125 APEC isolates is shown in Table 2. The *fimC* gene was the most prevalent, with a detection rate of 94.4%, followed by *hlyF* (80.0%), *iucD* (76.8%), *iroN* (74.4%), and *iss* (73.6%). Four genes encoding toxins (*lt*, *st*, *sxt1*, and *stx2*) were not detected in any isolate. The virulence genes *iucD*, *iroN*, *hlyF*, *vat*, and *iss* were significantly more often detected in isolates from chickens (84.4, 83.3, 87.5, 32.3, and 81.3%, respectively) than ducks (51.7, 44.8, 55.2, 3.4, and 48.3%, respectively) (*p* < 0.05). As shown in Table 3, MVS differed among the phylogenetic groups. The MVS of all isolates was 6.4, while the scores of chicken and duck isolates were 6.9 and 5.1, respectively. Phylogenetic group B2 harbored a multitude of VAGs and exhibited the highest score in both chicken (9.5) and duck (9.0) isolates, while groups A and E showed the lowest scores among chicken (A: 2.7, E: 4.6) and duck (A: 3.5, E: 3.0) isolates, respectively.

### 3.3. Antimicrobial Resistance Profiles of APEC Isolates

The MIC distribution and resistance of the 17 tested antimicrobials for the APEC isolates from chickens and ducks are shown in Table 4. Resistance to nalidixic acid was found most frequently (84.8%), followed by tetracycline (68.8%), sulfisoxazole (68.0%), and ampicillin (68.0%), whereas resistance was seen less frequently toward colistin (0.8%), amoxicillin/clavulanic acid (4.0%), azithromycin (4.0%), and cefoxitin (4.0%). Notably, the rates of resistance (18.8–75.0%) to ampicillin, ceftiofur, ceftriaxone, and gentamicin in chicken isolates were higher than those (0–44.8%) in duck isolates (*p* < 0.05). The MIC_90_ values for ceftiofur, ceftriaxone, and gentamicin of chicken isolates were higher, at >8, 64, and >16 μg/mL, respectively, than the 1, ≤0.25, and 1 μg/mL of duck isolates. MDR was found in most of the isolates (74.4%), and the isolates were most often resistant to 5 classes of antimicrobials (24.0%). The difference in MDR rate between chicken isolates (77.1%) and duck isolates (65.5%) was not statistically significant (Appendix A).

### 3.4. Prevalence of Antimicrobial Resistance Genes

The frequencies of 37 antimicrobial resistance genes in the APEC isolates are shown in Table 5. The antimicrobial resistance genes were screened for in the relevant antibiotic-resistant isolates. Among 85 APEC isolates resistant to ampicillin, most harbored genes encoding β-lactamase: *bla*_TEM_ (75 isolates, 88.2%), *bla*_CTX-M_ group I (7 isolates, 8.2%), and *bla*_CTX-M_ group IV (8 isolates, 9.4%). *bla*_CTX-M_ group I and *bla*_CTX-M_ group IV were only detected in chicken isolates. Most azithromycin-resistant isolates carried *mphA* alone (80.0%). Phenicol resistance genes were commonly detected in 53 isolates resistant to chloramphenicol: *cat* (47 isolates, 88.7%), *floR* (48 isolates, 90.6%), and *cmlA* (5 isolates, 9.4%). Streptomycin resistance genes, *strA-B* and *aadA*, were detected in 64 (87.7%) and 31 (42.5%) of 73 streptomycin-resistant isolates, respectively. Among 85 sulfisoxazole-resistant *E. coli*, 24 (28.2%) and 74 (87.1%) isolates carried *sul1* and *sul2*. Seventy-four (86.0%) of 86 tetracycline-resistant isolates harbored *tetA*, and 21 isolates (24.4%) carried *tetB*. Seven types of PMQR genes were investigated in 106 nalidixic acid-resistant isolates, but *qnrB*, *qnrS*, and *aac6′-1b-cr* were only found in 1 (0.9%), 11 (10.4%), and 2 (1.9%) isolates, respectively. One isolate was resistant to colistin (MIC 16 μg/mL), which only carried *mcr-1*. 

### 3.5. Comprehensive Characteristics as Measure of A Zoonotic Potential in Isolates Belonging to Phylogenetic Group B2

According to previous studies, susceptible uropathogenic *E. coli* (UPEC) isolates mainly belong to phylogenetic group B2. In this study, 22 (17.6%) of the 125 APEC isolates were classified as B2, and we used MLST to assess whether the B2 isolates were human-related sequence types (STs). In addition, virulence profiles and antimicrobial resistance were analyzed to identify the risk to public health posed by the B2 isolates. The comprehensive characteristics of the B2 isolates are shown in Figure 2. The 22 APEC isolates belonging to group B2 were mainly assigned to the O2 serogroup (54.5%, 12/22), followed by NT (22.7%, 5/22), O1 (18.1%, 4/22), and O18 (4.5%, 1/22). They were classified into 4 distinct STs: ST95 (77.3%, 17/22), ST140 (13.6%, 3/22), ST355 (4.5%, 1/22), and ST429 (4.5%, 1/22). The most frequent ST, ST95, is known to be associated with both humans and animals, according to the MLST database. The isolates identified as ST95 harbored more than 9 VAGs (common VAGs: *fimC*-*iroN*-*irp2*-*iucD*-*fyuA*-*tsh*-*hlyF*-*ompT*-*iss*), and 82.4% of them were MDR. Eight different resistance patterns were observed in these MDR isolates, the most common of which was ‘STR-FIS-SXT-AMP-CIP-NAL-ENR-TET’.

## 4. Discussion

Avian pathogenic *E. coli* (APEC) infections in poultry constitute a severe animal health problem and significant economic burden to farmers worldwide [1]. APEC is a subset of extraintestinal pathogenic *E. coli* (ExPEC), which also includes other pathotypes, such as uropathogenic *E. coli* (UPEC), neonatal meningitis *E. coli* (NMEC), and septicemia-causing *E. coli* (SEPEC) [1]. It has been reported that APEC is closely related to human ExPEC [7,58,59]. Therefore, the prevalence of colibacillosis in chickens and ducks in poultry farms causes two issues: an economic impact on poultry production and a zoonotic potential as a foodborne reservoir of ExPEC. Many studies analyzing the characteristics of APEC isolated from chickens and demonstrating similarities between APEC and human ExPEC have been performed in North America, Brazil, Thailand, Australia, and Europe [24,58]. Relatively few surveys of APEC isolated from ducks have been conducted worldwide [28,29]; in Korea, the overall characteristics and genetic relatedness of recent APEC isolates have hardly been investigated [32]. For this reason, we compared the characteristics of APEC isolates from chickens and ducks and assessed the zoonotic potential of the isolates.

APEC constitutes more than 180 serogroups, but serogroups O1, O2, O8, O35, and O78 are dominant in APEC throughout the world [7]. In Korea, between 2003 and 2005, the predominant serogroups were O78 (29.9%), O88 (8.2%), and O15 (7.2%) in chickens, and O78 (88.9%) in ducks [30]. In one study from 2018, O78 was the main serogroup identified in broiler chickens (20.3%) [32]. However, in the present study (conducted from 2018 to 2019), the predominant serogroup was O2 (12.5%), followed by O1 (6.3%), O45 (6.3%), and O78 (4.2%) in chickens, including broiler, layer, and breeder, whereas O78 was still predominant in ducks. The change in O-serogroup distribution is presumed to be dependent on various factors, such as the sampled farms and breeds of chickens, as well as the timing of the sampling. However, because O2 was prevalent in various breeds (broiler, layer, broiler breeder, and Korean native chicken) in the recent isolates in this study, the inclusion of the vaccine against O2 with the commercial vaccine against O78 currently available in Korea is recommended.

In addition to O-serogroups, *E. coli* strains are divided into eight phylogenetic groups (A, B1, B2, C, D, E, F, and clade 1) [8]. In previous studies, APEC isolates showed a diverse distribution of phylogenetic groups, including A, B1, B2, C, and F, depending on sampling areas and times [9,10,11]. In this work, group B1 (36.5%, 35/96) and group A (34.5%, 10/29) were predominant in the isolates from chickens and ducks, respectively. In prior studies, human ExPEC isolates causing urinary tract infections (UTIs), meningitis, and bacteremia were mainly assigned to phylogenetic group B2, followed by group D [9,16]. In the present study, group B2, which is closely related to human ExPEC, was the second most frequent group in chicken isolates, accounting for 20.8%, whereas only two duck isolates (6.9%) were assigned to group B2. Our results suggest that APEC isolates from chickens are a greater zoonotic risk to humans than APEC isolates from ducks.

The extraintestinal pathogenesis of APEC is facilitated by a broad range of virulence-associated genes [14]. In this study, among 14 tested VAGs, the frequencies of *fimC*, *iucD*, *iroN*, *hlyF*, and *iss* in chicken isolates were more than 80% (81.3–96.9%). Previous reports from Spain, USA, Denmark, and Korea also indicated similar frequencies of these genes [17,26,32,60]. In the duck isolates in this study, *fimC* was the most frequently detected VAGs, at 86.2%, while the other VAGs were found in 0–62.1% of samples. Similarly, Wang et al. [28] reported that duck isolates in China harbored *fimC* (94.1%), whereas the frequency of *iss* (97.2%) was higher than that (48.3%) in our duck isolates. Interestingly, the virulence genes *iucD*, *iroN*, *hlyF*, *iss*, and *vat* were detected significantly more often in isolates from chickens (84.4, 83.3, 87.5, 81.3, and 32.3%, respectively) than ducks (51.7, 44.8, 55.2, 48.3, and 3.4, respectively) (*p* < 0.05). Although it is difficult to draw concrete conclusions by comparing a few VAGs in chicken and duck isolates, our findings suggest that chicken isolates may have a higher virulence potential than duck isolates. In addition, the *iucD*, *hlyF*, and *iroN* genes have previously been described as being strongly associated with A/E-PEC, which are a group of isolates in phylogenetic branches of mixed APEC and human ExPEC strains [60]. The chicken-sourced APECs isolated in this study seem to be more strongly related to human ExPEC. However, there is a limit to the quality of these inferences because our survey included a number of small duck-sourced APEC isolates and a lack of investigation of other VAGs. Further comparative studies and investigation of other virulence factors are required. On the other hand, as shown in Table 3, we observed similar MVS between chicken and duck isolates and differences in MVS among phylogenetic groups. Similar to the results of El-Shaer et al. [33], in this study, group B2 showed the highest overall MVS as well as the highest score within each separate category of chicken and duck isolates. Saha et al. [27] also reported that group B2 isolates harbored all of the tested VAGs, while group A and B1 isolates possessed few VAGs. These results indirectly support the assertion made by previous authors that phylogenetic group B2 is the most virulent in most cases of ExPEC infections [20,61,62].

Antimicrobial treatment has been a cost-effective practice for reducing both the incidence and mortality rate of avian colibacillosis. However, overuse and misuse of antibiotics have provided selective pressure for the emergence of antimicrobial resistance strains and MDR strains, leading to therapy failure and potential economic losses in the worldwide poultry industry [19,20]. In the present study, almost 75% of the isolates were MDR, which is similar to reports from other countries, including Brazil (71.0%), Egypt (77.6%), and China (89.2%) [31,63,64]. By comparison with the MDR rate (86.2%) of Korean APEC isolates from 2003 to 2005 [31], our MDR rate was slightly lower, but it has remained high over a long period of time. The resistance to critically important antimicrobials (CIA) for human medicine, such as fluoroquinolone and third-generation cephalosporins, is an especially serious concern related to the poultry industry. More than 60% of our isolates were resistant to fluoroquinolones (ciprofloxacin and enrofloxacin), which is probably because fluoroquinolones have been used indiscriminately for treatment and prophylaxis of bacterial disease since their introduction into the poultry industry in 1990. In addition, the rate of resistance to third-generation cephalosporins (ceftiofur and ceftriaxone) in our chicken isolates had jumped to more than 18% compared to about 1% of the chicken isolates in a study conducted from 2003 to 2005 [31]. In addition, our chicken isolates were more resistant to third-generation cephalosporins than were our duck isolates (*p* < 0.05). The increased resistance to third-generation cephalosporins in recent chicken isolates is attributed to the fact that ceftiofur has been administered in ovo or by subcutaneous injection to 1-day-old chicks, together with Marek’s disease and IBD vaccination in Korea since 2002. The spread of these CIAs in poultry poses a public health threat because it is becoming more difficult to treat bacterial diseases in humans as antibiotic resistance is transmitted to humans through poultry products contaminated with antimicrobial-resistant strains [65]. Therefore, it is necessary that government regulators, veterinarians, and farmers in Korea address the growing problem of the use of these CIAs in the livestock industry to minimize the spread of antimicrobial resistance.

In addition to the spread of antimicrobial-resistant strains, antimicrobial resistance genes can be transferred and disseminated between food-producing animal and human pathogens, which is a public health concern around the world [65]. Thus, we also investigated the presence of antimicrobial resistance genes in the APEC isolates. Two groups of β-lactamase-encoding genes were identified in ampicillin-resistant isolates and *bla*_TEM,_ which only codes for narrow-spectrum β-lactamases that can inactivate penicillins and aminopenicillins, was the most prevalent. *bla*_TEM_ has also been detected in Japan, Australia, Egypt, and Korea [32,66,67,68]. CTX-M, an extended-spectrum β-lactamase, confers resistance to most β-lactam antimicrobials, including third-generation cephalosporins [69]. It can lead to an increase in the resistance to other antimicrobials via horizontal gene transfer [70]. In the present study, *bla*_CTX-M_ group I and *bla*_CTX-M_ group IV were also detected. In Sweden, Germany, U.K., and Korea, *bla*_CTX-M_ group I was previously reported in *E. coli* isolates from poultry, and *bla*_CTX-M_ group IV was reported in China [71]. Resistance to quinolones is mediated by chromosomal target site mutations and plasmid-mediated quinolone resistance (PMQR) mechanisms [71]. Several PMQR mechanisms have been identified, including those mediated by Qnr-like proteins (*qnrA*, *qnrB, qnrC*, *qnrD*, and *qnrS*), which protect DNA from quinolone binding; *aac(6′)-Ib-cr* acetyltransferase, which modifies fluoroquinolones; and active efflux pumps (*qepA*) [71]. In particular, the global dissemination of PMQR genes is a serious public health concern [71]. Our APEC isolates showed the highest antimicrobial resistance to nalidixic acid but carried few PMQR genes (*qnrB*, 0.9%; *qnrS*, 10.4%; *aac(6′)-Ib-cr*, 1.9%). However, these genes can be transferred to other pathogenic strains by conjugative plasmids, and the expression of these genes can increase the MICs of nalidixic acid and ciprofloxacin by two-fold to eight-fold and eight-fold to 32-fold, respectively [72]. Colistin has been approved for veterinary medicine for the prevention and treatment of disease in Korea [73]. In the current study, one of 125 isolates was resistant to colistin and carried *mcr-1*, colistin-resistance gene. This study is the first to report the detection of this gene in APEC in Korea. Colistin resistance has become a considerable concern due to transmission from animals to humans. In particular, colistin is a last-resort antibiotic in humans. The emergence of colistin resistance in MDR bacteria would compromise antibiotic treatment in humans. Our *mcr-1*-positive isolate was resistant to the largest number of antimicrobial classes (aminoglycosides-cephalosporins-folate pathway inhibitors-macrolides-β lactam-polymyxins-quinolones-tetracyclines) among all MDR isolates. For this reason, our detection of *mcr-1* in APEC poses two threats: poor antimicrobial treatment in poultry and the potential risk associated with consumer exposure to poultry products from the food chain.

As mentioned above, many researchers have suggested that APEC may pose a risk to public health for reasons such as sharing of various virulence-associated genes, transmission of antimicrobial resistance genes, resistance to the antibiotics needed for human medical care, and dissemination of MDR isolates [20,61,74]. In addition, according to recent reports, a subset of ExPEC strains from specific clonal groups, including ST95 and ST23, could cause diseases in both humans and chickens [20]. We confirmed, through MLST analysis, that ST95 was mainly distributed in our isolates belonging to the B2 group. Jorgensen et al. [60] found a large degree of genetic overlap and a wide dispersion of clones closely related to both APEC and human ExPEC belonging to ST95 through an in-depth whole-genome-based comparison study. These findings support the previous hypotheses that certain ST95 ExPECs either do not exhibit host specificity or have multiple host specificity, including humans and birds. In the present study, B2-ST95 APEC isolates showed higher MVS (9.2) than all isolates (6.4), and 82.4% of the isolates were MDR. In particular, 58.9% of these isolates were resistant to trimethoprim/sulfamethoxazole and fluoroquinolones (ciprofloxacin and enrofloxacin), which are used as first-line and second-line antibiotics, respectively. In Korea, B2-ST95 UPEC isolates are most commonly identified in urine samples of UTI patients [12], and most of these isolates are susceptible to tested antimicrobials. However, if the B2-ST95 APEC strains that were found to be MDR and resistant to CIA in the present study are passed to humans from farm to fork, difficulties with antibiotic treatment may result. Therefore, proactive control measures, such as sanitation management and vaccinations, must be taken at the poultry farm level to ensure animal welfare and public health.

## 5. Conclusions

This study compared the distributions of serogroups, phylogenetic groups, VAG patterns, and antimicrobial resistance between APEC isolates from chickens and ducks. Although the number of duck isolates in the present study was small, this type of comparative study on the characteristics of isolates from chickens and ducks is rarely done, so the information is quite valuable. Our APEC isolates from chickens showed significantly higher frequencies of some VAGs (*iucD*, *iroN*, *hlyF*, *iss*, and *vat*) and increased rates of resistance to third-generation cephalosporins (ceftiofur and ceftriaxone) compared to those from ducks (*p* < 0.05). The APEC isolates from chickens may have a higher virulence potential than the duck isolates. In addition, we found that 13.6% of all isolates were B2-ST95, which were identified as UTI-causing UPECs in previous studies. Our findings highlight the zoonotic potential and public health concerns related to these APEC. Poultry products could be a reservoir for bacteria in the same genetic lineage as human ExPEC and MDR strains and allow for the dissemination of antimicrobial resistance genes.

## Figures and Tables

**Figure 1 microorganisms-09-00946-f001:**
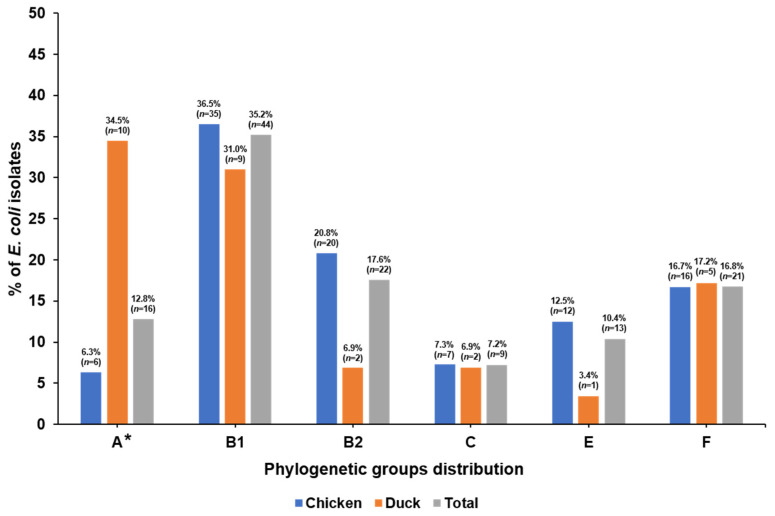
Distribution of phylogenetic groups among 125 APEC isolates. *: significant difference (*p* < 0.05) between chicken and duck isolates.

**Figure 2 microorganisms-09-00946-f002:**
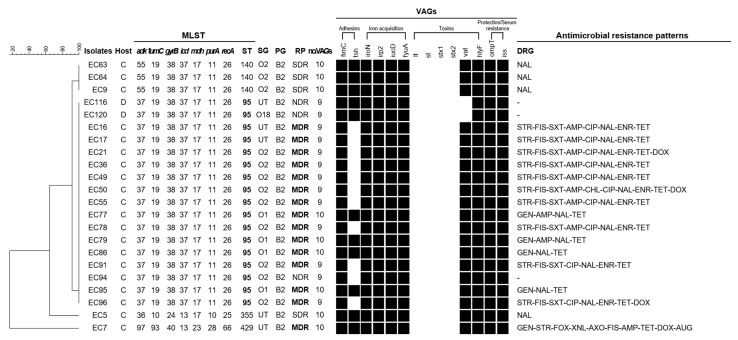
Characteristics and dendrogram based on Multi-locus sequence typing (MLST) among the 22 APEC isolates belonging to group B2. Phylogenetic tree based on the 7 concatenated gene sequences. Black squares: positive reaction; white squares: negative reaction; ST: sequence type; SG: serogroup; PG: phylogenetic group; RP: resistant phenotype; no.VAGs: number of virulence-associated genes; C: chicken; D: duck; NDR: non-drug resistant; SDR: non-multidrug resistant; MDR: multidrug resistant; AMP: ampicillin; AUG: amoxicillin/clavulanic acid; AXO: ceftriaxone; CHL: chloramphenicol; CIP: ciprofloxacin; DOX: doxycycline; ENR: enrofloxacin; FIS: sulfisoxazole; FOX: fefoxitin; GEN: gentamicin; NAL: nalidixic acid; STR: streptomycin; SXT: trimethoprim/sulfamethoxazole; TET: tetracycline; XNL: ceftiofur.

**Table 1 microorganisms-09-00946-t001:** O-serogroups of APEC isolates from chickens and ducks.

Chicken (*n* = 96)		Duck (*n* = 29)		Total (*n* = 125)	
No. of Isolates (%)	O-Serogroups	No. of Isolates (%)	O-Serogroups	No. of Isolates (%)	O-Serogroups
12 (12.5)	O2	3 (10.3)	O78	13 (10.4)	O2
6 (6.3)	O1, O45	2 (6.9)	O15, O25, O150	7 (5.6)	O1, O78
5 (5.2)	O88	1 (3.4)	O1, O2, O9, O18, O22, O24, O34, O39, O81, O84, O89, O129, O149, O156, O160, O81, O84, O89	6 (4.8)	O45
4 (4.2)	O78	5 (17.2)	Non-typable	5 (4.0)	O88
3 (3.1)	O102	-	-	4 (3.2)	O25
2 (2.1)	O8, O20, O24, O25, O177, O182	-	-	3 (2.4)	O24, O102
1 (1.0)	O5, O7, O21, O22, O29, O51, O55, O68, O115, O140, O143, O154, O166, O184	-	-	2 (1.6)	O8, O15, O20, O22, O150, O177, O182
34 (35.4)	Non-typable	-	-	1 (0.8)	O5, O7, O9, O18, O21, O29, O34, O39, O51, O55, O68, O81, O84, O89, O115, O129, O140, O143, O149, O154, O156, O160, O166, O184
-	-	-	-	39 (31.2)	Non-typable

**Table 2 microorganisms-09-00946-t002:** Frequency of virulence-associated genes in APEC isolates from chickens and ducks.

Virulence-Associated Gene	No. (%) of Positive Isolates		
	Chicken (*n* = 96)	Duck (*n* = 29)	Total (*n* = 125)
Adhesin			
*fimC*	93 (96.9)	25 (86.2)	118 (94.4)
*tsh*	46 (47.9)	15 (51.7)	61 (48.8)
Iron acquisition			
*iucD* *	81 (84.4)	15 (51.7)	96 (76.8)
*iroN* *	80 (83.3)	13 (44.8)	93 (74.4)
*irp2*	51 (53.1)	17 (58.6)	68 (54.4)
*fyuA*	46 (47.9)	13 (44.8)	59 (47.2)
Toxins			
*hlyF* *	84 (87.5)	16 (55.2)	100 (80.0)
*vat* *	31 (32.3)	1 (3.4)	32 (25.6)
*lt*	0 (0.0)	0 (0.0)	0 (0.0)
*st*	0 (0.0)	0 (0.0)	0 (0.0)
*stx1A*	0 (0.0)	0 (0.0)	0 (0.0)
*stx2A*	0 (0.0)	0 (0.0)	0 (0.0)
Protectins/serum resistance			
*iss* *	78 (81.3)	14 (48.3)	92 (73.6)
*ompT*	67 (69.8)	18 (62.1)	85 (68.0)

* Significant difference (*p* < 0.05) between chicken and duck isolates.

**Table 3 microorganisms-09-00946-t003:** Correlation between virulence-associated genes and phylogenetic groups among APEC isolates from chickens and ducks according to mean virulence score (MVS).

Phylogenetic Group	MVS ^a^ (No. of Isolates)		
	Chicken Isolates (*n* = 96)	Duck Isolates (*n* = 29)	All Isolates (*n* = 125)
Total	6.9 (96)	5.1 (29)	6.4 (125)
A	2.7 (6)	3.5 (10)	3.2 (16)
B1	5.8 (35)	3.7 (9)	5.3 (44)
B2	9.5 (20)	9.0 (2)	9.4 (22)
C	8.3 (7)	9.0 (2)	8.4 (9)
E	4.6 (12)	3.0 (1)	4.5 (13)
F	8.6 (16)	8.0 (5)	8.4 (21)

^a^ Mean virulence score (the sum of all VAGs detected in isolates in a given group/the number of isolates in that group).

**Table 4 microorganisms-09-00946-t004:** Antimicrobial susceptibility of APEC isolates from chickens and ducks.

Antibiotics	Break Point (μg/mL)	Chicken (*n* = 96)					Duck (*n* = 29)				Total (*n* = 125)		
		MIC Range (μg/mL)	MIC_50_ ^1^ (μg/mL)	MIC_90_ ^2^ (μg/mL)	R ^3^ (%)	MIC Range (μg/mL)	MIC_50_ (μg/mL)	MIC_90_ (μg/mL)	R (%)	MIC Range (μg/mL)	MIC_50_ (μg/mL)	MIC_90_ (μg/mL)	R (%)
Gentamicin	≥16 ^a^	0.5–16	1	>16	26.0 *	0.5–2	1	1	0.0	0.5–16	1	>16	20.0
Streptomycin	≥32 ^b^	4–64	>64	>64	62.5	8–64	16	>64	44.8	4–64	>64	>64	58.4
Ampicillin	>32 ^a^	≤1–32	>32	>32	75.0 *	2–32	4	>32	44.8	≤1–32	>32	>32	68.0
Amoxicillin/ Clavulanic Acid	≥32/16 ^a^	2–32	8	16	5.2	2–16	8	16	0.0	2–32	8	16	4.0
Cefoxitin	≥32 ^a^	2–32	4	8	5.2	2–16	4	8	0.0	2–32	4	8	4.0
Ceftiofur	≥8 ^b^	0.25–8	0.5	>8	18.8 *	0.25–1	0.5	1	0.0	0.25–8	0.5	>8	14.4
Ceftriaxone	≥4 ^a^	≤0.25–64	≤0.25	64	19.8 *	≤0.25	≤0.25	≤0.25	0.0	≤0.25–64	≤0.25	16	15.2
Sulfisoxazole	>512 ^a^	≤16–256	>256	>256	71.9	≤16–256	>256	>256	55.2	≤16–256	>256	>256	68.0
Trimethoprim/ Sulfamethoxazole	≥4/76 ^a^	≤0.12–4	0.5	>4	46.9	≤0.12–4	>4	>4	51.7	≤0.12–4	0.5	>4	48.0
Azithromycin	≥32 ^a^	2–16	4	16	3.1	2–16	4	>16	6.9	2–16	4	16	4.0
Chloramphenicol	≥32 ^a^	≤2–32	8	>32	38.5	4–32	32	>32	51.7	≤2–32	8	>32	41.6
Colistin	≥16 ^c^	0.5–16	0.5	0.5	1.0	0.5–1	0.5	0.5	0.0	0.5–16	0.5	0.5	0.8
Ciprofloxacin	≥1 ^a^	≤0.015–4	4	>4	64.6	≤0.015–4	2	>4	55.2	≤0.015–4	4	>4	62.4
Enrofloxacin	≥2 ^d^	≤0.12–32	8	32	63.5	≤0.12–32	4	>32	58.6	≤0.12–32	8	32	62.4
Nalidixic Acid	≥32 ^a^	1–32	>32	>32	85.6	2–32	>32	>32	82.8	1–32	>32	>32	84.8
Doxycycline	≥16 ^a^	0.5–64	8	64	45.8	1–64	16	64	51.7	0.5–64	8	64	47.2
Tetracycline	≥16 ^a^	≤4–32	>32	>32	70.8	≤4–32	>32	>32	62.1	≤4–32	>32	>32	68.8

^1^ The lowest concentration of the antimicrobials at which 50% of the isolates were inhibited. ^2^ The lowest concentration of the antimicrobials at which 90% of the isolates were inhibited. ^3^ Antimicrobial resistance frequency.^a^ Clinical Laboratory Standards Institute (CLSI), M100, 2018; ^b^ CDC. National Antimicrobial Resistance Monitoring System (NARMS), 2018; ^c^ European Committee on Antimicrobial Susceptibility Testing (EUCAST), 2018; ^d^ Clinical Laboratory Standards Institute (CLSI), M100, 2015; * The frequency of antimicrobial resistance in chicken isolates was significantly greater than in duck isolates (*p* < 0.05).

**Table 5 microorganisms-09-00946-t005:** Distribution of antimicrobial resistance genes among APEC isolates from chickens and ducks.

Antibiotics	Antimicrobial Class	Chicken					Duck				Total		
		No. of Resistance Isolates	Associated Genes	No. of Positive Isolates	(%)	No. of Resistance Isolates	Associated Genes	No. of Positive Isolates	(%)	No. of Resistance Isolates	Associated Genes	No. of Positive Isolates	(%)
Ampicillin	β-lactam	72 (72.5%)	*bla* _TEM_	64	88.9	13 (44.8%)	*bla* _TEM_	11	84.6	85 (68.0%)	*bla* _TEM_	75	88.2
			*bla*_CTX-M_ group Ⅰ	7	9.7		*bla*_CTX-M_ group Ⅰ	0	0.0		*bla*_CTX-M_ group Ⅰ	7	8.2
			*bla*_CTX-M_ group Ⅳ	8	11.1		*bla*_CTX-M_ group Ⅳ	0	0.0		*bla*_CTX-M_ group Ⅳ	8	9.4
Azithromycin	Macrolides	3(3.1%)	*mphA*	3	100.0	2(6.9%)	*mphA*	1	50.0	5 (4.0%)	*mphA*	4	80.0
Chloramphenicol	Phenicols	38 (39.2%)	*cmlA*	2	5.3	15 (51.7%)	*clmA*	3	20.0	53 (42.4%)	*clmA*	5	9.4
			*Cat*	36	94.7		*Cat*	11	73.3		*Cat*	47	88.7
			*floR*	35	92.1		*floR*	13	86.7		*floR*	48	90.6
Colistin	Polymyxins	1(1.0%)	*mcr1*	1	100.0	0(0.0%)	*mcr1*	0	0.0	1 (0.8%)	*mcr1*	1	100.0
Nalidixic acid	Quinolones	82 (85.4%)	*qnrB*	0	0.0	24 (82.8%)	*qnrB*	1	4.2	106(84.8%)	*qnrB*	1	0.9
			*qnrS*	7	8.5		*qnrS*	4	16.7		*qnrS*	11	10.4
			*aac6’-1b-cr*	2	2.4		*aac6’-1b-cr*	0	0.0		*aac6’-1b-cr*	2	1.9
Streptomycin	Aminoglycosides	60 (61.9%)	*strA-B*	54	90.0	13 (44.8%)	*strA-B*	10	76.9	73 (58.4%)	*strA-B*	64	87.7
			*aadA*	25	41.7		*aadA*	6	46.2		*aadA*	31	42.5
Sulfisoxazole	Folate pathway inhibitors	69 (71.9%)	*sul1*	19	27.5	16 (55.2%)	*sul1*	5	31.3	85 (68.0%)	*sul1*	24	28.2
			*sul2*	62	89.9		*sul2*	12	75.0		*sul2*	74	87.1
Tetracycline	Tetracyclines	68 (70.8%)	*tetA*	61	89.7	18 (62.1%)	*tetA*	13	72.2	86 (68.8%)	*tetA*	74	86.0
			*tetB*	17	25.0		*tetB*	4	22.2		*tetB*	21	24.4

Antimicrobial resistance genes (β-lactam: *bla_SHV_*; macrolides: *mphB*, *ermA*, *ermB*, *ermC*, *ereA*, *ereB*, *mefA*, *msrA*; polymyxins: *mcr2*, *mcr3*, *mrc4*, *mcr5*; quinolones: *qnrA*, *qnrC*, *qnrD*, *qepA*; tetracyclines: *tetC*, *tetD*, *tetE*, *tetG*), which were not detected in both chicken and duck isolates are not shown in this table.

## Data Availability

Not applicable.

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
