# Peer review of "Comparative Characteristics and Zoonotic Potential of Avian Pathogenic Escherichia coli (APEC) Isolates from Chicken and Duck in South Korea"

_microorganisms, 2021, doi:10.3390/microorganisms9050946_

Round 1
Reviewer 1 Report
The investigation reported is a standard investigation of isolates obtained from APEC in chicken and duck in S. Korea. The investigation is well done and presented, however, the question is why we need it. What is new compared to previous studies and what is new from S. Korea? What is new compared to other international studies? The information from duck may be new and by comparing the dataset to chicken it is possible to rate the potential virulence and public health importance of the duck isolates. The majority of the investigation is still from chicken and we get little information of pathology or even the type of poultry analysed. "Chicken" can both be broilers and layers. Also the type of ducks is not mentioned. With the right perspective and comparison the data might be viewed in a new light, else it is difficult to see the new information in this paper and why it needs to be published.
Author Response
The authors thank the reviewers for their constructive comments. Our responses to each of the comments are provided below.
Reviewer #1
1. The investigation is well done and presented, however, the question is why we need it. What is new compared to previous studies and what is new from S. Korea?What is new compared to other international studies?
Response: Few surveys of APEC isolated from ducks have been conducted in Korea and other countries. Some reports investigated some characteristics of APEC isolated in 1999-2008 and 1993, and so it is difficult to understand the characteristics of recent APEC isolates from ducks. Therefore, we isolated APEC from ducks with colibacillosis that occurred in Korea between 2018 and 2019, and investigate the overall characteristics of the isolates, such as serotypes, phylogenetic groups, antimicrobial resistance, and the presence of antimicrobial resistance genes and virulence associated genes. This study was able to present the latest information on APEC isolates from ducks which has not been available in Korea and other countries. In addition, we analyzed the zoonotic potential of B2-ST95 isolates from chickens and ducks by investigating the number of virulence associated genes and antimicrobial resistance. We think that our data could be the basis for control measures on farms and further research.
2.The information from duck may be new and by comparing the dataset to chicken it is possible to rate the potential virulence and public health importance of the duck isolates. The majority of the investigation is still from chicken and we get little information of pathology of even the type of poultry analyzed.
Response: when we compared APEC isolates from chickens and ducks, the prevalent serotypes and phylogenetic groups were different, and compared to duck isolates, chicken isolates were significantly more resistant to critically important antimicrobials (CIAs) such as third-generation cephalosporins, and more specific virulence associated genes were detected. Therefore, we drew the inference that the APEC isolates from chickens may have a higher virulence potential than the duck isolates. Although the virulence potential of duck isolates itself could not be raised in this study, we believe that comparison between chicken and duck isolates helped to highlight the relative health risk of chicken isolates.
3.“Chicken” can both be broilers and layers. Also, the type of ducks is not mentioned.
Response: All breeds of chickens (layer, broiler, broiler breeder, and Korean native chicken) and ducks (broiler duck and breeder duck) were described in Supplementary table 3.
Reviewer 2 Report
The paper presents the still up-to-date topic concerning poultry colibacillosis as a disease caused by pathogenic strains of Escherichia coli, including pathogenic avian Escherichia coli (APEC). The article is well written and worth publishing. The article can be accepted as sent.
Author Response
The authors thank the reviewers for their constructive comments. Our responses to each of the comments are provided below.
1.The paper presents the still up-to-date topic concerning poultry colibacillosis as a disease caused by pathogenic strains of Escherichia coli, including pathogenic avian Escherichia coli (APEC). The article is well written and worth publishing. The article can be accepted as sent.
Response: Thank you for the encouraging comment. We hope that this study will help people understand the characteristics of APEC prevalent in Korea and the potential health risks of the specific type. Also, we hope that our overall investigation we conducted will be reproducible in other countries, so that the characteristics of the APEC isolates can be analyzed, and proper management can be carried out on the farm.